# Audiologist's Perspective in Auditory Rehabilitation: Implications for Ethical Conduct and Decision-Making in Portugal

Tatiana Marques [1,2,3,*] , Margarida Silvestre [4,5] , Bárbara Santa Rosa [4] and António Miguéis [6]

1 CIBIT—Coimbra Institute for Biomedical Imaging and Translational Research, University of Coimbra, 3004-531 Coimbra, Portugal
2 Faculty of Medicine, University of Porto, 4099-002 Porto, Portugal
3 Audiology, Physiotherapy and Environmental Health Department, Coimbra Health School, Polytechnic Institute of Coimbra, 3004-531 Coimbra, Portugal
4 Institute of Bioethics, Faculty of Medicine, University of Coimbra, 3004-531 Coimbra, Portugal; msilvestre@fmed.uc.pt (M.S.); santa.rosa.b@gmail.com (B.S.R.)
5 Reproductive Medicine Department, Coimbra Hospital and University Centre, 3004-531 Coimbra, Portugal
6 Clinic of Otorhinolaryngology, Faculty of Medicine, University of Coimbra, 3004-531 Coimbra, Portugal; amigueis@fmed.uc.pt
* Correspondence: tatiana.marques@estescoimbra.pt; Tel.: +351-239-543-152

**Abstract:** Ethical standards in audiology have been continuously improved and discussed, leading to the elaboration of specific regulatory guidelines for the profession. However, in the field of auditory rehabilitation, audiologists are still faced with circumstances that question their ethical principles, usually associated with the support of the hearing aids industry. The study explores the decision-making process and ethical concerns in auditory rehabilitation as they relate to the practice of audiology in Portugal. An online questionnaire constructed by the authors was used and sent to the email addresses of a list of audiologists, registered with the Portuguese Association of Audiologists. The questionnaire was answered by 93 audiologists with clinical experience in auditory rehabilitation for more than one year. The collected data demonstrated that audiometric results and clinical experience are the most important factors for decision-making in auditory rehabilitation practice. Moreover, incentives from the employers or manufacturers were identified as the main cause of ethical dilemmas. This study highlights the ethical concerns regarding the clinical practice of auditory rehabilitation in Portugal, revealing that the decision-making process is complex and, specifically in this field, the current practice may not be adequate for effective compliance with professional ethical standards.

**Keywords:** audiology; auditory rehabilitation; decision-making; ethic; ethical dilemmas

## 1. Introduction

Decision-making in audiology is an important factor in the success of rehabilitation, serving as a strategic resource of information and influence on the patient [1]. Indeed, several studies pointed to a patient-centered approach for improving and facilitating this process based on a shared decision-making perspective, in which patient choices and decisions about hearing care are reflected in the intervention.

Given the complex process and challenges of auditory rehabilitation, the decision-making process is fundamental in situations that generate conflict between personal and professional integrity [2,3]. Therefore, this process should be based on defining whether the situation constitutes an ethical dilemma, i.e., conditions that confront moral standards, beliefs, and values, which contribute to challenging audiologist decisions in their practice.

Further, to decide whether a situation should be considered an ethical dilemma, the audiologist should gather and take into consideration all the circumstances while exploring different perspectives on the problem [4].

The term ethics refers to judgements not exclusively about what is right or wrong, but also about what is praiseworthy or blameworthy [4,5]. A code of ethics for professional associations is pivotal in maintaining the ethical standards of audiological practice [6–8]. The current code of ethics in Portugal was published in 2005 by the Portuguese Association of Audiologists (APtA), which asserts ethical guidelines [9]. In fact, worldwide, the principles announced in codes of etics' emphasize that the practice of audiology should be performed with honesty in clinical practice, maintaining elevated values of competence and being only responsible for services that serve the health interests of patients' health [9,10]. However, codes of ethics comprise only general concepts, without no or only a brief overview of guidelines in ethical decision-making. Thus, ethical codes fail to address ethical issues in audiology practices, specifically in the field of auditory rehabilitation.

According to the health survey of the Instituto Nacional de Saúde Doutor Ricardo Jorge in 2015 [11], it is estimated that 1.6. million people in Portugal have hearing loss, however, only 2.7% use hearing aids (HAs), and of these 50% have hearing complaints. In Portugal, HAs can be provided by the National Health Service (NHS) or purchased by the patients themselves in private hearing centers. In both situations, they are prescribed by an otorhinolaryngologist, although in the NHS the total cost of HAs is subsidized. However, this process is time-consuming and implies that, after the appointment to deliver the hearing aids at the hospital, the fitting and adjustment of HAs is carried out in a private center specialized in HAs, which was selected as a supplier by the hospital [12]. Conversely, when the patients purchase the HAs for themselves, they can choose the brand and type of hearing aid, as well as the supplier. However, in both situations, some ethical dilemmas may arise in the clinical practice of the audiologists. For example, in the case of HAs subsidized by NHS, audiologists are often faced with situations in which the HA is not suitable for that patient due to the progression of hearing loss. Sometimes, the last clinical evaluation was performed a long time ago, and the process is so time-consuming, taking more than 2 years, that by the time the patient receives the device, it is no longer what the audiologist would advise. In fact, after specialist prescription of hearing aids, there is a long waiting time due to the lack of funding that hospitals or specialized centers have per year for hearing aids, leading to extensive waiting lists. After waiting for so long for the HAs, it is difficult for the patient if the audiologist cancels the process, even when the patients know that the benefit with that device will be lower. This situation leads to an ethical dilemma for these professionals, between ensuring that the patient receives the most adequate HA for their hearing or keeping the now outdated device purchased by the hospital. Providing the most adequate HA may involve starting over again after two years of waiting, while keeping the HA provided by the hospital may ask more effort from the audiologist to provide enough functional gain and the best possible performance with the outdated device, nevertheless leading to less benefit for the patient compared to starting the fitting procedure with a new device. In addition, when the patient chooses to buy the HA on his own, there is no reimbursement from the government, and the high cost of these devices, associated with the patients' financial issues, limits the options for fitting and counseling of the audiologist.

Moreover, in the specific field of auditory rehabilitation, audiology is commonly sponsored by the industry, with professionals working close with hearing device manufacturers, which leads to conflicts of interest related to their funding, products and personnel. For example, HA audiologists face incentive opportunities and pressures to adjust hearing devices from specific brands or manufacturers, which induce misconduct and challenges ethical standards in their daily practice [13]. In addition, these ethical issues in auditory rehabilitation increase patients' distrust in audiologists, as well as in audiology as a profession. Patients can be affected if audiologists have minimal knowledge of ethics and, as a result, have difficulty identifying ethical tensions. In Portugal, audiologists working in

auditory rehabilitation belong mainly to the private sector, which can lead to a higher risk of exposure to ethical dilemmas related to financial incentives or pressure from employers, as well as more complex decision-making processes. Thus, audiologists' opinions and behaviors are crucial to improving HA adoption rates. However, the clinical practice of the audiologists can be challenged with situations that question their ethical perception, leading to the misuse of principles and, consequently, unexpected behaviors in disagreement with ethical standards [14].

Comparing hearing care across the world, Manchaiah, Tomé, Duckens, Harn and Ganesan assume that clinical practice in the field of auditory rehabilitation is quite similar [15]. However, their findings suggested that the clinical practice setting and reimbursement of hearing healthcare systems vary considerably across countries, due to audiologists' specific differences in audiology education and training. Previously, similar results were described by Goulios and Pattuzi, who conducted a survey of audiologists' practice from 62 countries, verifying differences in the clinical practice of audiologists between developing and developed countries related to government funding and training of audiologists [16]. In fact, Li et al. conducted a review of auditory rehabilitation programs in China and showed that the Chinese government has expanded and specialized its auditory rehabilitation services, which has contributed to reducing the social and economic burden of hearing impairment [17]. In contrast, in developing countries, HAs are not a priority for health systems, leading to inadequate provision of these devices and insufficient audiologists to assure their needs [16]. Therefore, differing models of healthcare provision may contribute to the differences and similarities observed in the conduct of audiologists. Furthermore, the decision-making process and the patient approach can influence the adoption rate of HAs, as described in previous surveys [18–20].

Cho et al. observed that the most common barriers to acceptance and adoption of HAs are not only financial constraints, but also the progression of hearing loss and the audiologists' counseling [18]. In fact, communication deficits increase proportionally to the progression of HL, leading to a better acceptance of HAs. Furthermore, it seems that HA adoption is higher if advised by an audiologist in a private center than in a hospital due to the patients' expectations that another type of treatment will be possible in the hospital. Otherwise, the different approaches of audiologists have a significant influence on their relationship with the patient, revealing that patients prefer a patient-centered approach in their experiences and preferences.

Manchaiah, Gomersall, Tomé, Ahmadi and Krishna examined audiologists' preferences in Portugal, India and Iran, showing that there are some differences in the preferences of audiologists from these countries. Particularly, audiologists in Portugal seem to favor a patient-centered approach compared to audiologists in India and Iran, even after the effects of age and professional experience have been accounted for. However, the authors suggested that the decision-making context influences the approach assumed by the hearing care professional [21]. Thus, if the focus is on diagnostic, a medical approach is chosen, otherwise, in rehabilitation, a model based on shared decisions between the patient and audiologist is the preference across countries. Therefore, this study aims to: (a) identify which sources of information the audiologists use in clinical practice; (b) assess the perceived difficulty and self-confidence in decision-making; and (c) identify sources of ethical problems that audiologists face in auditory rehabilitation.

## 2. Materials and Methods

Approximately 170 emails were sent to all the audiologists registered with APtA. The principal investigator provided an explanation of the survey and a link to an author-constructed online questionnaire. Audiologists were chosen according to the following inclusion criteria: (i) undergraduate or master's degree in audiology; (ii) clinical practice in auditory rehabilitation for >1 year. Demographic information was collected regarding age, gender, marital status, a higher degree of education, region of practice, and experience. The questionnaire addressed topics, such as the participants' perception of their clinical

practice and professional experience, relations(hip) with the HA industry, a vision of the ethical implications of the audiology–industry interface, or ethical tensions that arise in practicing at this interface. Thus, categories of ethical dilemmas included academic aspects (i.e., ethics in research and in education), compliance with regulatory guidelines, interactions with colleagues or supervisors, clinical aspects, interactions between the family and caregiver, and financial incentives. The categories used for decision-making were similar, corresponding to the clients' goals and preferences, discussions with the colleagues or supervisors, trial and error, previous experience, and the manufacturer's or employer's incentives (please see the questionnaire in the Supplementary Materials). The categories used in the questionnaire were based on a similar survey of ethical dilemmas in audiology developed by Callahan et al. [4].

Statistical analysis was performed using the SPSS software for Windows, version 27 (IBM SPSS, Chicago, IL, USA). Descriptive statistics were performed to measure the central tendency, including mean, median and standard deviation for continuous variables, or N and percentage (%) for discrete variables. These features quantitively describe the collection of information, allowing the interpretation of the most evident ethical issues that arise in auditory rehabilitation. Statistical analysis between professional experience and self-reported difficulty in decision-making, as well as between professional experience and the influence of a supervisor was completed using the Kruskal–Wallis test, followed by post hoc analysis with the Mann–Whitney test to compare categories according to professional experience. Finally, Bonferroni correction was applied to correct *p*-values according to the number of additional analyses. Consequently, a *p*-value < 0.0083 after the Bonferroni correction was considered statistically significant.

## 3. Results

The sample included 93 audiologists who were characterized by age, gender, education, marital status, region of practice, and experience (see Table 1 for participant descriptions). The mean age of the participants was 32.35 SD 7.80 years old and most participants who completed the questionnaire have their practice in the north (38.7%) and centre (35.5%) regions of the country. An undergraduate degree in audiology (81.7%) is the most occurring education.

**Table 1.** Participant descriptions.

|  |  | N (%) or Mean (SD) | Maximum | Minimum |
|---|---|---|---|---|
|  | **Age** | **32.35 (7.80)** | **60** | **23** |
| Gender | Male | 20 (21.5%) | - | - |
|  | Female | 73 (78.5%) | - | - |
| Marital Status | Married | 38 (40.9%) | - | - |
|  | Cohabitation | 19 (20%) | - | - |
|  | Not married | 36 (38.7%) | - | - |
| Education | Undergraduate | 76 (81.7%) | - | - |
|  | Master | 16 (17.2%) | - | - |
|  | Doctoral | 1 (1.1%) |  |  |
| Region of practice | North | 36 (38.7%) | - | - |
|  | Center | 33 (35.5%) | - | - |
|  | Lisbon and Tejo Valley | 18 (19.4%) | - | - |
|  | Alentejo | 1 (1.1%) | - | - |
|  | Algarve | 2 (2.2%) | - | - |
|  | Archipelago of Madeira | 2 (2.2%) | - | - |
|  | Archipelago of Açores | 1 (1.1%) | - | - |
| Experience (years) | 1 to 4 | 29 (31.2%) | - | - |
|  | 5 to 9 | 35 (37.6%) | - | - |
|  | More than 10 | 29 (31.2%) | - | - |

Abbreviations: N, number; %, percentage; SD, standard deviation.

First, it was observed that professionals' decisions were mainly guided by the results of audiometric tests (75.3%) and clinical experience (21.5%). In contrast, a patient's and colleague's opinion, manufacturers and professional guidelines were graded as the least crucial sources for decision-making.

Thereafter, participants were asked what type of information they would use during the HA fitting process, namely, selecting the most appropriate HA, if they were presented in a scenario with conflicting information or when there is no obvious decision. Participants' responses indicated that 63.4% considered the client's goals and preferences in their decision-making and 20.4% preferred discussing the case with colleagues. However, 1.1% of the 93 respondents mentioned the use of trial and error (as shown in Table 2). Interestingly, in these difficult cases, only 10.8% rely on previous experience. Likewise, only 4.3% used the discussion with the supervisor in these cases.

**Table 2.** Categorization for the question "What type of information would use in a scenario where the optimal solution is not clear or there is conflicting information or contraindication for the patients' auditory rehabilitation?".

| Category | N (%) |
|---|---|
| Clients' goals | 59 (63.4%) |
| Discussion with colleagues | 19 (20.4%) |
| Previous experience | 10 (10.8%) |
| Discussion with supervisors | 4 (4.3%) |
| Trial and error | 1 (1.1%) |
| Manufactures or employer financial incentives | 0 (0%) |

Abbreviations: N, number; %, percentage; SD, standard deviation.

Answers related to compliance with professional regulatory guidelines were also relevant, with 62.1% of the audiologists not using any of the available regulatory guidelines. However, the remaining responses from participants who indicated using regulatory guidelines varied and suggested the use of a combination of several guidelines, namely using a combination of APtA, American Speech-Language-Hearing Association, and British Society of Audiology guidelines. Interestingly, audiologists with longer clinical practice tend to report following professional guidelines more often.

Results of self-perceived difficulty showed that difficulty in decision-making was rated as slightly for 43% of respondents. As expected, significant differences were observed when comparing self-perceived difficulty with professional experience ($p$-value < 0.001). Therefore, respondents with less experience (1 to 4 years) have more difficulty in making clinical decisions during their practice when the information or clinical findings are unclear (see Figure 1), when compared to professionals with 5 to 9 years, or more than 10 years of professional practice ($p$-value < 0.001). However, when comparing audiologists with 5 to 9 years of professional practice with audiologists with more than 10 years, no statistical differences were found ($p$-value = 0.873).

In addition, when asked to rate self-perceived confidence in decision-making, audiologists mainly reported very confident (51.6%) to complete confident (40.9%), as can be observed in Table 3.

**Table 3.** Rate for self-perceived difficulty in decision-making.

| Category | N (%) |
|---|---|
| Not at all confident | 0 (0%) |
| Slightly confident | 3 (3.2%) |
| Moderately confident | 4 (4.3%) |
| Very confident | 48 (51.6%) |
| Complete confident | 38 (40.9%) |

Abbreviations: N, number; %, percentage; SD, standard deviation.

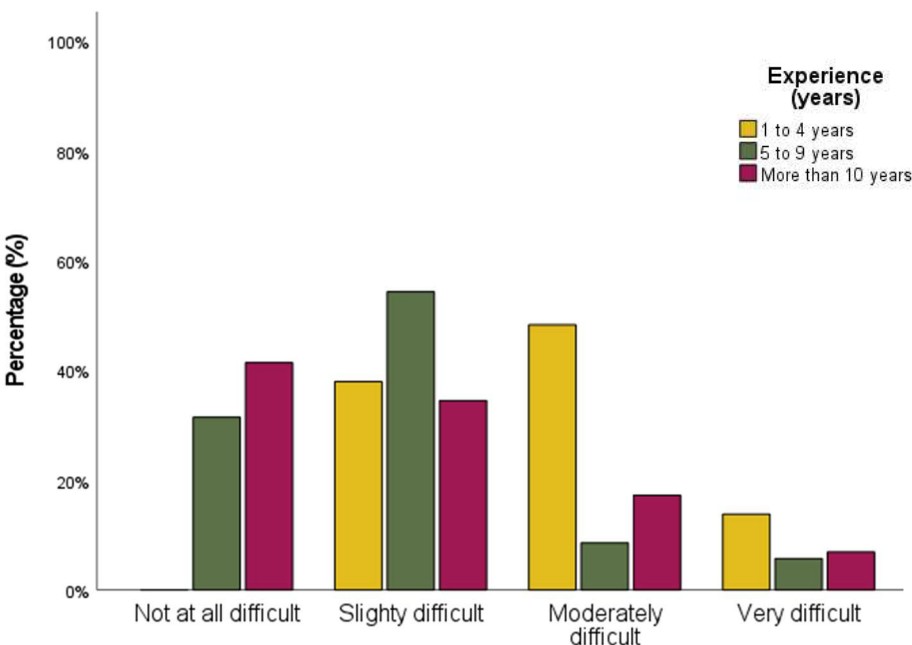

**Figure 1.** Bar graphs showing self-perceived difficulty in decision-making according to experience (years).

When answering the question, "In your experience, which of the following alternatives has caused you ethical concerns in clinical practise?", respondents selected financial incentives as the main cause of ethical concern (32.3%), making this the most frequently reported category, followed by interactions between the family and caregiver (11.8%) and interactions with colleagues or with the supervisor (7.5%). However, other factors were appointed as susceptible to create ethical dilemmas during an audiologist's practice, namely, compliance with regulatory guidelines, clinical and academic aspects, or a combination of the categories mentioned. In addition, 92% of the interviewees stated that they received financial incentives, with 30% receiving them according to the number of HAs fitted and the other 20% varied with the technology of HAs. Moreover, 23.7% of respondents reported unethical conduct, describing that they gave incomplete or incorrect information to the patient. Findings also identified ethical dilemmas related to noncompliance with treatment recommendations or non-acceptance of the diagnosis by family members/caregivers, discouraging the patient from maintaining follow-ups or performing additional audiological evaluations when requested. A range of responses pointed to colleagues/supervisors' interactions as responsible for ethical dilemmas due to superior interference that calls into question professional autonomy. Testimonial statements from respondents relate experiences in which supervisors gave inaccurate information to patients, committed insurance fraud, or were even incentivized to work with unqualified personnel who had not completed any audiology course (15.1%).

The influence of the supervisor on decision-making was also questioned (see Figure 2) and the results showed differences related to professional experience (*p*-value < 0.001).

A Mann–Whitney test with Bonferroni correction identified that audiologists with less experience (1 to 4 years) had a significantly different rank (*p*-value < 0.001) from respondents with 5 or more years of professional practice as HA audiologists, showing that the supervisor has a higher influence on professionals with 1 to 4 years of practice.

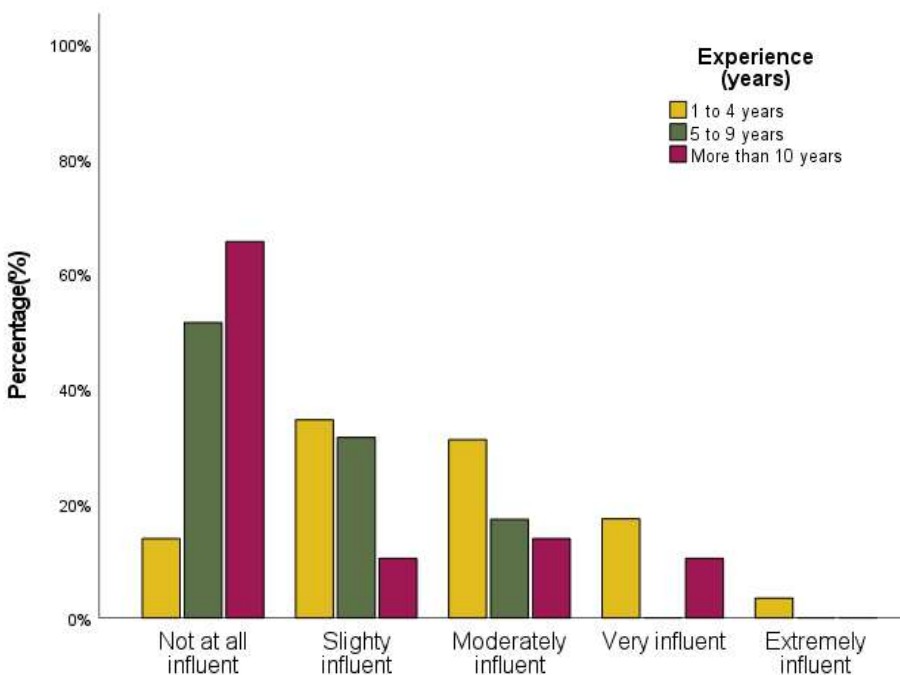

**Figure 2.** Bar graph showing self-perceived supervisor's influence on decision-making according to experience (years).

## 4. Discussion

This study highlights the ethical dilemmas that arise in the practice of auditory rehabilitation in Portugal, as well as the factors that contribute to the decision-making of audiologists. The results show that the most decisive factor for decision-making is based on clinical findings, with 75.3% of respondents considering audiometric results as the most important factor. Our findings corroborate the ones of Boisvert et al. [13], who describe that the results of the audiometric test are the most important factor to the audiologist practitioner, as well as clinical experience. The opposite was observed in our study regarding the goals and preferences of patients, which was ranked as one of the least information sources for decision-making. Previously, Manchaiah, Gomersall, Tomé, Ahmadi and Krishna suggested that the decision-making approach is different according to the clinical context, concluding that in rehabilitation in Portugal a patient-centered approach is privileged, and their opinion and preferences are valued [21]. Similarly, from an examination of our results, audiologists' decision-making factors vary depending on the situation. However, we hypothesized that a patient-centered approach is essentially adopted if there is an ethical dilemma for the audiologist. In fact, respondents' most decisive factor was audiometric results, but when asked about which source of information to use to decide in a scenario where there is conflicting information, the major preference was the clients' goals and preferences (63.4%), which suggests that when faced with an ethical dilemma, Portuguese audiologists choose a patient-centered method. Therefore, it seems that the use of information related to patient factors is incorporated into their daily practice when there was no clear solution for the case. In addition, the experience of colleagues is also considered when faced with an ethical dilemma, with 20.4% of respondents reporting that they prefer to discuss the case with colleagues.

Effectively, Boisvert et al. [13] suggested that information related to patient characteristics should be incorporated into clinical decision-making to promote a patient-centered model that requires a balance between the recognition of vital information and the understanding of its consistency and pertinence to the clinical issue, engaging the patient and sharing decision-making. Therefore, we suggest a more deliberate and active involvement of patients in decision-making, which would increase clinical accountability. Effectively,

the Salzburg Statement on shared decision-making reinforces the importance of shared decisions with patients, respecting their goals and preferences, and ensuring that accurate information is given about the process, its benefits and risks, as well as its uncertainties [22]. Involvement of the patient in decisions related to auditory rehabilitation can lead the patient to recognize their hearing handicap and to improve the adoption rate of HAs.

Otherwise, discussing the case with colleagues when confronted with complex or inconsistent information was identified as important for audiologists to make their clinical decision. In fact, Crukley, Dundas, McCrury, Meston and Ng [23] described a dialogue between five audiologists about the challenges for audiology in clinical practice, education and research, who conclude that a truly collaborative practice needs to be achieved, suggesting that interprofessional collaboration should occur to assist patient needs, especially when those needs are complex. Active involvement of patients can improve the rehabilitation process and make it more consistent, as well as increase its effectiveness. Nonetheless, there are guidelines that were developed to minimize errors and standardize procedures, as well as to guarantee ethical conduct in clinical practice due to the challenge that audiologists face daily [12,14]. One of the aims of regulatory guidelines for professional associations is precisely to provide regular standards of audiological clinical assistance by offering training and support to their members, facilitating a robust and consistent ethos.

In our study, respondents' answers revealed low adhesion to the use of these guidelines, which can lead to notable inconsistencies between the suggested practice and the existing one. For example, compared to previous studies, which consistently ranked practice guidelines as an extremely important tool to decision-making, we observed a low use of these guidelines, and were directly related to the age and experience of the interviewed audiologists [4–6]. These findings suggest that clinical practice may be developing without following rigorous guidelines or engaging patients in clinical decisions and that the local protocols are often based on consensus within a specific clinical institution. If guidelines were used frequently, they would have the potential benefit of decreasing variability in the methods of practice and reducing error rates [5]. Among audiologists who reported not using clinical regulatory guidelines, this may be partly due to the private organizational culture. In fact, private audiology services present organizational policies and procedures that are developed to promote productivity, creating specific protocols for implementation in the clinical practice of their audiologists, which can lead to the use of guidelines less than audiologists who work in public services. In addition, supervisors are seen as coaches, possibly being the support that these audiologists consider necessary to maintain consistent standards of clinical service [13]. Therefore, in audiology, private practice may be necessary to reevaluate how clinical decisions are made, such as the existing ethical tension and the effects on patients.

The results also indicated that financial incentives are one of the major causes of ethical dilemmas for the audiologists interviewed. It's overwhelming that 92% of respondents receive these incentives, considering that this is ranked as one of the factors that undermines their ethical integrity. Based on respondents' comments, the audiology-industry relationship and employer incentives can be an issue, as it can affect an audiologist's judgment in clinical decisions, leading to inadequate amplification or unethical practice and noncompliance with professional guidelines. Therefore, everyday ethical challenges in their practice can lead to ethical distress related to the inability to endorse their own moral principles. That is, financial incentives have their advantages, however, they can change the entire course of action and conduct of the professional, since it generates ethical dilemmas. Over the years, continued exposure to ethical tensions can manifest as emotional distress, physiological stress, fatigue and exhaustion, and withdrawal from patients [24,25]. Furthermore, this behavior and professional (mis)conduct can lead to public distrust in auditory rehabilitation, as well as in audiology as a profession. For example, Ng et al. [3] suggest that patients' self-perception and well-being can be influenced by non-verbal communication, social and cultural implications, affecting how patients perceive themselves and the audiologists, and their impairment and disability in the larger community. Further,

50% of the respondents admitted to receiving financial incentives that varied with the number of hearing aids fitted or the technology of the device, which can contribute to exposing audiologists to pressures to increase the number of fitted patients, even when it is not clear whether this is the best option for the patient, which can result in inadequate amplification. However, collaboration with the hearing aid industry provides important training opportunities for auditory rehabilitation, making it crucial to the clinical practice of audiologists. This collaboration must be carried out in a selfless and knowledgeable way to guarantee benefit and avoid manipulation or dishonest promises to patients [23].

Conversely, other factors can raise ethical dilemmas, namely when a more expensive HA provides better benefits to the patient; however, the patient's low financial resources can influence the audiologists' choice of the device [4]. Given that audiologists want to provide the best technology available, and at the same time avoid balance-billing the patient, the decision-making process can be extremely difficult and contribute to an ethical dilemma [4,17,26]. Thus, to improve the ethical decision-making process, it is mandatory that the audiologists recognize the ethical dilemmas that they are facing, analyze the relevant facts, as well as examine the values in conflict to start the process. In addition, continuing education in audiology should incorporate formations in ethical decision-making.

Moreover, in Portugal, audiologists are represented by APtA, a not-for-profit organization without legal authority and, therefore, without regulatory and supervisory competencies. Currently, regulatory functions are the responsibility of the Ministry of Health, rather than being reallocated to this organization, which maintains an oversight of the knowledge, skills and practice of audiology as a profession, in addition to being an advisory member of the Ministry of Health on issues related to the profession [20]. Our results reflect this situation, with distrust of professional bodies and low adherence to regulatory guidelines of professional bodies. However, the absence of mandatory guidelines can be responsible for notable discrepancies between recommended best practices and current practices, so it is urgent to reassess how clinical decisions are made in audiology, to promote the continuous evolution of ethical standards for the practice of the profession. Finally, supervisors or directors must not allow or even carry out actions that breach accepted and ethical practices, or even discriminate professional integrity.

## 5. Limitations

This study has some limitations, namely the response rate of 50% that was obtained for this web-based questionnaire. Further, a potential sample bias, resulting from the low uptake from audiologists practicing in Alentejo, Algarve, Archipelago of Madeira, and the Azores, could influence the results. In addition, aspects, such as audiologist practice in diagnosis simultaneously with auditory rehabilitation were not controlled for but may have contributed to the differences in audiologists' preferences in decision-making. Other factors related to demographic data collection (e.g., nationality, religion, type of employment, etc.) may have some influence on recognizing an ethical dilemma, as well as identifying and managing ethical conflicts, and should be included in future research.

## 6. Conclusions and Recommendations

Professionals' decisions in audiology in Portugal are mainly guided by results of audiometric tests and clinical experience. Compliance with professional regulatory guidelines is very poor. Clients' goals and preferences are used by 63% of audiologists in decision-making. Audiologists with less experience (1 to 4 years) find decision-making more difficult than audiologists with more experience. Self-perceived confidence, however, is high for all levels of experience. Various sources of ethical dilemmas have been identified, such as financial incentives, interactions with family, and interactions with colleagues or supervisors. Almost all respondents indicate that they receive financial incentives and 27.3% admit to unethical conduct. Audiologists with less experience perceive more influence of supervisors on decision-making than audiologists with more experience.

This shows that the ethics of audiology as a profession and as a science are complex and remarkably context-dependent. Ethical dilemmas can appear at any time, so a broad understanding of practical, ethical and legal concerns should proceed and follow the auditory rehabilitation in such difficult circumstances. A complete set of decision-making orientations does not exist, therefore audiologists must be able to recognize and resolve ethical dilemmas, particularly concerning finance, coworker/supervisor interactions, compliance with regulatory guidelines, and family/caregiver interactions. In addition, professional bodies, employers and audiology courses should highlight these areas to furnish possible solutions and enhance ethical decision-making.

**Supplementary Materials:** The following supporting information can be downloaded at: https://www.mdpi.com/article/10.3390/audiolres12020020/s1.

**Author Contributions:** Conceptualization, T.M.; methodology, T.M.; formal analysis, T.M.; investigation, T.M. and A.M.; writing—original draft preparation, T.M.; writing—review and editing, A.M., M.S. and B.S.R.; visualization, A.M., M.S. and B.S.R.; supervision, A.M., M.S. and B.S.R.; project administration, T.M. All authors have read and agreed to the published version of the manuscript.

**Funding:** This research received no external funding.

**Institutional Review Board Statement:** The study was conducted in accordance with the Declaration of Helsinki and approved by the Ethics Committee of the Faculty of Medicine, University of Coimbra (Approval number CE-108/2020) in September 2020.

**Informed Consent Statement:** Informed consent was obtained from all subjects involved in the study.

**Data Availability Statement:** Not applicable.

**Acknowledgments:** The authors would like to gratefully acknowledge APtA for generously contributing to this research, disseminating the questionnaire by email.

**Conflicts of Interest:** The authors declare no conflict of interest.

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
