# Peer review of "Audiologist’s Perspective in Auditory Rehabilitation: Implications for Ethical Conduct and Decision-Making in Portugal"

_audiolres, doi:10.3390/audiolres12020020_

Round 1
Reviewer 1 Report
Dear Authors,
First of all, I would like to congratulate you with the paper you have submitted. It addresses an important topic, namely ethics in Audiological decision making. You have developed a questionnaire based on one found in the literature and used it to make an online survey that was responded to by 93 audiologists in Portugal. You present the results and discuss implications.
I feel that there are some points for improvement that will strengthen the paper and that will make in even more interesting for an international public of Audiologists. My overall comment is that providing more detailed information of how hearing care is organized in Portugal will give the reader more background to understand why this topic is important. Furthermore, adding literature about shared decision making may provide a clearer framework for the discussion.
Below I will present my recommendations, section by section.
- Introduction
In the first paragraph you mention some literature about ethical standards in Audiology as an introduction of the topic. There is more literature, however that can provide a framework for discussing your results.
For example, the call to action of the 17th World Health Assembly, 13 may 2017, Agenda item 15.8. Amongst others, it urges member states to:
- to integrate strategies for ear and hearing care within the framework of their primary health care systems, under the umbrella of universal health coverage, by such means as raising awareness at all levels and building political commitment and intersectoral collaboration;
- to establish suitable training programmes for the development of human resources in the field of ear and hearing care;
- to improve access to affordable, cost-effective, high-quality, assistive hearing technologies and products, including hearing aids, cochlear implants and other assistive devices, as part of universal health coverage, taking into account the delivery capacity of health care systems in an equitable and sustainable manner;
- to work towards the attainment of Sustainable Development Goal 3 (Ensure healthy lives and promote well-being for all at all ages) and Goal 4 (Ensure inclusive and equitable quality education and promote lifelong learning opportunities for all) in the 2030 Agenda for Sustainable Development, with special reference to people with hearing loss;
It would strengthen the paper to discuss in the introduction how audiological care is integrated into the Portugese health care system. The WHO World Report on Hearing, published in March 2021, also may provide a framework for the discussion.
Recommendation 1
Please provide in the introduction an overview of how audiological care is integrated in the Portuguese health care system and possibly discuss how this may be different to other international contexts.
Another topic that should be discussed in the introduction is recent literature on shared decision making. At least The Salzburg Statement on Shared Decision Making and two important papers on the subject are worth mentioning.
The Salzburg Statement on Shared Decision Making. (2011). Zeitschrift Für Evidenz, Fortbildung Und Qualität Im Gesundheitswesen, 105(4), 325. https://doi.org/10.1016/s1865-9217(11)00138-3
Stiggelbout, A. M., Van Der Weijden, T., De Wit, M. P. T., Frosch, D., Légaré, F., Montori, V. M., Trevena, L., & Elwyn, G. (2012). Shared decision making: Really putting patients at the centre of healthcare. BMJ (Online), 344(7842), 1–6. https://doi.org/10.1136/bmj.e256
Elwyn, G., Frosch, D., Thomson, R., Joseph-Williams, N., Lloyd, A., Kinnersley, P., Cording, E., Tomson, D., Dodd, C., Rollnick, S., Edwards, A., & Barry, M. (2012). Shared decision making: A model for clinical practice. Journal of General Internal Medicine, 27(10), 1361–1367. https://doi.org/10.1007/s11606-012-2077-6
Recommendation 2
Please provide in the introduction an overview of ethical and practical aspects of shared decision making.
Finally, recommendation 1 and 2 can be related in the following way. The way audiological care is organized in Portugal may create barriers for correct implementation of shared decision making, resulting in ethical dilemmas.
Recommendation 3
Please add the discussion on page 2 lines 56-64 expand upon the specific barriers and facilitators of shared decision making resulting from the organization of audiological care and training of Audiologists in Portugal.
Recommendation 4
With recommendations 3, a hypothesis of the sources of ethical problems may be established and tested. It would strengthen the paper to rewrite study aim (c), page 2 line 67 and 68: The hypothesis will be tested that …. will be sources of ethical problems that audiologists mention …
- Material and Methods
Recommendation 5
Page 2, lines 72-73, please provide a more detailed description of construction of the questionnaire, specifically mentioning discussing why some categories were chosen and others not. It would strengthen the paper further is these choices can be related to recommendations 3 and 4.
Recommendation 6
Please provide the Portuguese and English translated versions of the questionnaire in an appendix.
Recommendation 7
Page 2 , lines 80-85, Please provide more specific information on how descriptive statistics were presented. For example, there is a difference in reporting continuous variables versus discrete variables. Also, normal distribution is required for a mean and standard deviation to be a meaningful representation of central tendency.
- Results
Recommencation 8
Page 2, lines 87-92, please provide information about how well the respondents represent the general population of Audiologists in Portugal. If the responders are not a good representation, this should be mentioned as a weak point of the study in the discussion.
Recommendation 9
Page 2, lines 87-92, please provide information about nonresponders. If there is a significant difference in characteristics between responders and nonresponders, this should be mentioned as a weak point of the study in the discussion.
Recommendation 10
Please provide the results of the questionnaire in a table. For example, with each question in a different row and with columns showing the answers for categories of respondents according to experience. This will allow easy comparison of the results.
Recommendation 11
Page 3, lines 101-104, it is unclear to me what is meant by this sentence.
Recommendation 12
Page 4, table 2 may be incorporated in the table of recommendation 10.
Recommendation 13
Page 4, lines 135-137, this sentence seems more appropriate for the discussion.
Recommendation 14
Page 4, line 141, please check grammar.
Recommendation 15
Page 4, line 146, please check grammar.
Recommendation 16
Page 4, lines 149-150, please check grammar.
- Discussion
Recommendation 17
The discussion should start with discussing the results in relation to the study aims and the framework constructed in the introduction. Next the results can be compared to those found in the literature. Finally, recommendation for improvement could be given. Also, strong and weak points of the study should be discussed, such as possible sources of bias.
Success with further preparations of the Manuscript.
Kind Regards,
Alex Hoetink
Author Response
Dear reviewer,
Thank you for taking the time to review our work. In the review it is stated that the manuscript needs changes in order to strengthen the paper, to provide more detailed information about hearing care in Portugal and add literature about decision-making. We have addressed the suggested revisions throughout the body of the article. In order to make the paper easier to follow we have reorganized results and added comments. We have modified the manuscript according to the comments below. We have also updated the literature search and the statistical analyses to maintain the use of discrete variables.
Replies to specific comments are given below.
Recommendation 1, 2 and 3
Response - We agree that an overview of hearing care in Portugal should be integrated, so we have added some literature on hearing care in Portugal in Introduction (page 1, lines 35-45), as well as on decision-making (page 2, lines 92-105). The Salzburg Statement on Shared Decision Making was used in Discussion (page 8, lines 266-269). Lastly, examples of barriers associated with the organization of hearing care in Portugal have been added to the paper (page 2, lines 46-53).
Recommendation 4
Response – Our study is exploratory, therefore, we intend to identify sources of ethical concerns and obtain criteria to develop an approach to the problem. Thus, no hypothesis was previously stablished for the study.
Recommendation 5
Response – We added more information about questionnaire (page 3, lines 136-135).
Recommendation 6
Response – We added both versions of the questionnaire (Appendix A and B).
Recommendation 7
Response – We agree that there is a difference in reporting continuous and discrete variables, so we performed a new statistical analysis, more suitable for discrete variables, using the Kruskall-wallis test and rewrote the results (page 4 to 8).
Recommendation 8
Response – We added a subtitle with limitations, which include this topic. This study has a response rate of 50%, however, this percentage is from all the audiologists registed at APtA, which means that not all of them have practice in auditory rehabilitation, and probably it is a good representation of audiologists that work in this field (page 9, lines 347-356).
Recommendation 9
Response – We do not have information on non-respondents, due to the confidentiality of APtA data. However, all questions were mandatory, except for the testimonies.
Recommendation 10 and 12
Response – We chose to present the results in a graph to facilitate the visualization of the data. The analysis was always carried out in relation to the total sample and exclusively to assess the supervisor's influence and the self-perceived difficulty in decision making, the analysis of the groups according to age was performed, which will be reflected in figures 1 and 2 (page 9 and 10).
Recommendation 11
Response – We corrected the sentence (page 4, lines 168-169)
Recommendation 13
Response – Sentence moved to the discussion (page 8, lines 302-305)
Recommendation 14, 15 and 16
Response – Grammar checked
Recommendation 17
Response - We agreed with the recommendation and the discussion was restructured (page 7 to 9).
Reviewer 2 Report
This manuscript would be better categorized as a "brief report" instead of a full research article given the content and page size. It summarizes an ethics survey among audiologists in Portugal and concluded that the current practice in terms of compliance with ethical standards needs improvement in the audiology field in Portugal.
The topic is current and important. There are several areas that need improvement.
- The brief report can benefit from including the questionnaire as supplementary material.
- Exactly what factors were included in the ANOVA analysis needs to be specified. What were the categories according to professional experience? What scores were compared? Why was one-way ANOVA appropriate? For instance, did the data points meet the assumptions for ANOVA? Could years of practice be treated as continuous variable and generalized mixed effects model be used here?
- Were similar surveys conducted before? Please consider the following studies:
- Audiologists’ preferences for patient-centredness: a cross-sectional questionnaire study of cross-cultural differences and similarities among professionals in Portugal, India and Iran | BMJ Open
- Full article: A qualitative investigation of decision making during help-seeking for adult hearing loss (tandfonline.com)
- Thieme E-Journals - Journal of the American Academy of Audiology / Abstract (thieme-connect.com)
- Thieme E-Journals - Journal of the American Academy of Audiology / Full Text (thieme-connect.com)
- Patient-centred hearing care in Malaysia: what do audiologists prefer and to what extent is it implemented in practice?: Speech, Language and Hearing: Vol 21, No 3 (tandfonline.com)
- Full article: Patterns in the social representation of “hearing loss” across countries: how do demographic factors influence this representation? (tandfonline.com)
- Preferences to Patient-Centeredness in Pre-Service Speech and Hearing Sciences Students: A Cross-Sectional Study (nih.gov)
- Development of the concept of patient-centredness – A systematic review - ScienceDirect
- When comparing across cultures (see some studies above), was there something unique or specific to the audiology profession in Portugal? What are the global trends?
- What are the limitations of the survey results? What can be improved?
Author Response
Dear reviewer,
Thank you for taking the time to review our work. Replies to specific comments are given below.
- We added Portuguese and English versions of the questionnaire (Appendix A and B).
- We performed a new statistical analysis, more suitable for discrete variables, using the Kruskall-wallis test and rewrote the results (page 4 to 8).
- The suggested studies are indeed important and were included in the literature used for this exploratory study, however this is the first survey that analyzes the perspective of audiologists in relation to the ethical challenges they face in their clinical practice in Portugal. In other countries, such as the USA, a study was previously carried out (Callahan et al.), whose questionnaire even served as the basis for the construction of the one used in this research. However, they are countries with different cultural realities and health care, therefore it is important to better understand the reality of Audiology practice in the world and to improve and approximate procedures.
- There are some particularities related to the specific area of ​​auditory rehabilitation in Portugal. For example, in Portugal, auditory rehabilitation is almost exclusively performed in private centers, but these particularities are described in the Introduction (page 1, lines 35-45). The global trend is precisely the opposite, as reported at the 17th World Health Assembly, in which the need to improve access to free and universal hearing care is highlighted.
- We added a subtitle with limitations, which include this topic. This study has a response rate of 50%, however, this percentage is from all the audiologists registed at APtA, which means that not all of them have practice in auditory rehabilitation, and probably it is a good representation of audiologists that work in this field (page 9, lines 347-356).
Round 2
Reviewer 1 Report
Dear Authors,
Please find attached my second review report.
Best Wishes.

Reviewer 2 Report
The authors have addressed my concerns in the revision. I recommend acceptance.
Author Response
Thank you for the time and effort dedicated to providing your valuable feedback on this manuscript.
Round 3
Reviewer 1 Report
Dear authors,
The paper has improved much in this version and I congratulate you with this achievement. I still have some suggestions for further improvement, however, after which it will be fit for publication in my humble opinion.
Line 37: I think appointed to should either read pointed to …. for improving or pointed out …. for improving.
Line 43: conditions that confronts … should be conditions that confront ….
Line 50: is a pivotal should read is pivotal.
Line 62: complains should be complaints.
Line 66: Maybe it adds to clarity to add latter: However, this latter process …
Line 80: I suggest to change even knowing in even when the patient knows …
Lines 81-84: I would suggest to rephrase this sentence as follows: … , between ensuring that the patient receives the most adequate HA for their hearing or keeping the now outdated device purchased by the hospital. Providing the most adequate HA may involve starting over again after two years of waiting, while keeping the hearing aid provided by the hospital may ask more effort from the audiologist to provide enough functional gain and the best possible performance with the outdated device, nevertheless leading to less benefit for the patient compared to starting the fitting procedure with a new device.
Line 126: HA instead of HAs?
Line 128: influence on instead of influence in?
Line 129-130: … patients prefer an approach centered in… I think this should read … patients prefer a patient-centered approach in …
Line 146: I suggest to remove A sampling of. Mails are either send to a sample of the audiologists or to all audiologists.
Line 148: I think … accordingly to … should either read … accordingly …. or … according to …
Lines 170-172: I suppose post-hoc testing was performed with the Mann Whitney U test to test differences in self-reported difficulty in decision making and influence of supervisor between the categories 1 to 4 years and 5 to 9 years, 1 to 4 years and More than 10 years and 5 to 9 years and More than 10 years. This means that 6 pos-hoc comparisons where made. Benferroni correction implies using a value for significance of p < 0.05/6 = 0.0083. If this is correct, I would mention it in the methods section. Please also provide information about the software package and version that was used for the statistical analyses.
Line 184: … patient’s … instead of patient?
Line 188: I think what is meant with the remaining responses … are the responses of the participants that did indicate that they use regulatory guidelines … Please clarify this in the sentence.
Line 199: rely in the previous should read rely on previous. Likewise, … and 4.3% on discussion …
Line 205-208: I still find this part confusing. From the text it is unclear what percentages are meant. As I understand it, the first sentence refers to total percentage for all categories for the question about self-perceived confidence. The two percentages do not add up to 100% however, so there is data that is not presented.
Next the phrase According to these results, suggests that the following information presented still concerns the question about self-perceived confidence, while in fact it is about self-perceived difficulty. Also, different terms for level of difficulty in the text (slight) and figure (not at all difficult), cause confusion. Studying appendix A to clarify does not help either, as the order of questions in the appendix and in the results section is reversed.
I would therefore suggest to present the results in the same order as in the questionnaire. So first present the results about self-perceived difficulty (figure 1) and then about self-perceived confidence (preferably also in a figure with the same categories, even if the number of responses are small for some categories). Also use the same terms in the text as in the figures.
The results about influence of supervisor can then be presented in figure 3. (Note that this question is repeated in the English version of the questionnaire).
The advantage is that results for the questions with a rating (1 – 5) are all presented in the same way. This increases consistency and makes the results easier to understand in relation to appendix A.
Line 233: pointed to … instead of pointed …
Line 237: incentive work … should be incentive to work
Line 238: complete should be completed
Line 244: The post hoc analysis is probably a Mann Whitney U test with Benferroni correction for multiple testing. See methods section.
Line 251: I would strongly suggest that terms that are used in the questionnaire are used consistently in the text. For example, the first option does not correspond with the text of the question in Appendix A. There the term decisive factor is used. I would therefore suggest to rewrite the sentence as follows: The results show that the most decisive factor for decision-making is …
Line 256: least reliable sources for decision making. In appendix A the question is phrased differently. Possibly you intend to ask with the question: what information do you look for? This is something else than reliability. I think it is important to check if the English question is the right translation of the Portuguese question. If the Portuguese question asks about reliability, this should be reflected in the translation. Now participants could have had in mind information that is most easily available, instead of information that is reliable.
That is why it is important to check whether terms are used consistently in the translated questions and in the text of the paper.
Line 259: proximity to the patient is privileged … I’m not sure what is meant by this. Is it a privilege that audiologists are close to the patients? Or do you want to express that a patient-centered approach is preferred by audiologists in Portugal?
Line 263: Here again first option is used instead of decisive factor.
Line 267-269: now suddenly discussing the case with colleagues is introduced. It is not clear to me how this is related to information related to patient factors.
Line 275: involvement of patients instead of from?
Line 278: accurate information is given/provided
Line 279: Involvement of the patient?
Line 303: Possibly you mean local protocols instead of guidelines. Also remove and
Line 306-307: It is unclear to me how the requirement of reevaluating clinical decisions may result in not using guidelines. Please explain.
Line 319-320: Do you have a reference to support this statement?
Line 331-343: This paragraph needs some attention regarding grammar and readability. I would suggest shortening the sentences to increase readability.
Line 349-350: Please specify which results reflect the distrust of professional bodies. Not following guidelines by itself may not reflect distrust. Lack of time or recourses can also be a reason to not follow guide lines.
Line 355-366: … actions that breach …
Lines 370-378: The conclusion section could reflect more of the outcomes presented in the paper. It now reads more like a recommendation without direct link to the results.
As a suggestion:
Rename the section Conclusions and Recommendations and start with:
Professional decisions in audiology in Portugal are mainly guided by results of audiometric tests and clinical experience. Compliance to professional regulatory guidelines is very poor. Clients’ goals and preferences are used by 63% of audiologist in decision making. Audiologists with less experience (1 to 4 years) find decision-making more difficult than audiologists with more experience. Self-perceived confidence, however, is high for all levels of experience. Various sources of ethical dilemmas have been identified such as financial incentives, interactions with family and caregivers and interactions with colleagues or supervisors. Almost all respondents indicate that they receive financial incentives and 23.7% admit to unethical conduct. Audiologists with less experience perceive more influence of supervisors on decision-making than audiologists with more experience.
This shows that ethics of audiology as a profession and as a science … (continue with lines 370-378)
Line 373: I’m not sure Otherwise, is the right word to use. Maybe it can be omitted.
Line 374: therefore, … is maybe more appropriate than however,
Line 375: particularly concerning finance? …
Best wishes!

Author Response
Dear reviewer,
Thank you for the time and effort dedicated to providing your valuable comments of this manuscript.
General Response: Thank you for your suggestions. We agree with them and incorporate them throughout the manuscript. In addition, we modified the organization of paragraphs and corrected grammatical errors as suggested.
Comment: Lines 170-172: I suppose post-hoc testing was performed with the Mann Whitney U test to test differences in self-reported difficulty in decision making and influence of supervisor between the categories 1 to 4 years and 5 to 9 years, 1 to 4 years and More than 10 years and 5 to 9 years and More than 10 years. This means that 6 pos-hoc comparisons where made. Bonferroni correction implies using a value for significance of p < 0.05/6 = 0.0083. If this is correct, I would mention it in the methods section. Please also provide information about the software package and version that was used for the statistical analyses.
Response: Thanks for your suggestion. We have incorporated more detailed information about the statistical analysis (lines 165-176).
Comment: Line 259: proximity to the patient is privileged … I’m not sure what is meant by this. Is it a privilege that audiologists are close to the patients? Or do you want to express that a patient-centered approach is preferred by audiologists in Portugal?
Response: Thanks for pointing this out. We want to refer to a patient-centered approach, so the information has been modified in the manuscript (lines 269-270).
Comment: Lines 63: Line 205-208: I still find this part confusing. From the text it is unclear what percentages are meant. As I understand it, the first sentence refers to total percentage for all categories for the question about self-perceived confidence. The two percentages do not add up to 100% however, so there is data that is not presented.
Next the phrase According to these results, suggests that the following information presented still concerns the question about self-perceived confidence, while in fact it is about self-perceived difficulty. Also, different terms for level of difficulty in the text (slight) and figure (not at all difficult), cause confusion. Studying appendix A to clarify does not help either, as the order of questions in the appendix and in the results section is reversed.
I would therefore suggest to present the results in the same order as in the questionnaire. So first present the results about self-perceived difficulty (figure 1) and then about self-perceived confidence (preferably also in a figure with the same categories, even if the number of responses are small for some categories). Also use the same terms in the text as in the figures.
The results about influence of supervisor can then be presented in figure 3. (Note that this question is repeated in the English version of the questionnaire).
The advantage is that results for the questions with a rating (1 – 5) are all presented in the same way. This increases consistency and makes the results easier to understand in relation to appendix A.
Response: We agreed and have changed paragraphs organization, as well as a new table (table 3) was added to clarify the total percentage for all categories related to self-confidence. The same terms were used in text and in the figures or tables.
Comment: Line 306-307: It is unclear to me how the requirement of reevaluating clinical decisions may result in not using guidelines. Please explain.
Response: We have included clearer information in the text, explaining that private Audiology services have specific protocols for implementation in the clinical practice of their audiologists, which can lead to use guidelines less than audiologists who work in public services. In addition, we suggest that it is necessary to reassess how clinical decisions are made in the private practice of Audiology due to the low use of guidelines, but also the influence that supervisors can have (Lines 319-325).
Comment: Line 319-320: Do you have a reference to support this statement?
Response: References have been added to the text (Lines 339, 471-474).
We look forward to hearing from you in due time regarding our submission and respond to any further questions and comments you may have.
Thank you for your consideration!
Sincerely,
Tatiana Marques
